# Improving Durability of Dye-Based Polarizing Films Using Novel Reactive Dyes as Dichroic Materials

**DOI:** 10.3390/polym15224365

**Published:** 2023-11-09

**Authors:** Young Do Kim, Jea Uk Lee, Jung Jin Lee

**Affiliations:** 1Samsung Display Co., Ltd., Cheonan 31086, Republic of Korea; youngdo.k@daum.net; 2Department of Advanced Materials Engineering for Information and Electronics, Integrated Education Institute for Frontier Science and Technology (BK21 Four), Kyung Hee University, 1732 Deogyeong-daero, Giheung-gu, Yongin 17104, Republic of Korea; 3Department of Fiber System Engineering, Dankook University, Yongin 16890, Republic of Korea

**Keywords:** poly(vinyl alcohol), polarizing film, dye-based polarizing film, dichroic materials, reactive dyes, direct dyes

## Abstract

Iodine is commonly used as a dichroic material in polarizing films, while dichroic dyes are employed when high heat resistance is necessary. Direct dyes, which can be applied to poly(vinyl alcohol) (PVA) in an acidic environment, are the most popular; however, their hydrogen bonding interaction with the PVA chain can weaken in high-humidity conditions, leading to a potential change in color value or polarization property. Reactive dyes offer a promising alternative for use in high-humidity environments. In this study, five novel reactive dyes were synthesized and used to prepare dye-based polarizing films. The dichroic ratios, order parameters, and transition moments of the reactive dyes were calculated and compared to those of corresponding direct dyes. Molecular orbital calculations indicated minimal effects on the optical anisotropy of the polarizing films due to the transition moments of the reactive dyes remaining close to their molecular axes. As a result, the dichroic ratios of the polarizing films were mainly dependent on the orientation of the dyes towards their stretching direction, and showed a similar behavior compared to direct dyes. Under high-temperature and high-humidity conditions, the polarizing film using reactive dyes showed superior durability compared to the direct dye-based film.

## 1. Introduction

Polarizing films (PFs), used to control the transmittance of light in liquid crystal displays and reduce reflections in OLED panels, play a crucial role as optical components for both LCD and OLED displays, respectively. In general, polarizing films use iodine as a polarizing material. However, due to the sublimation of iodine at over 90 °C, light leakage of display with the iodine-based polarizing film occurs at that temperature. In order to overcome this problem, dichroic dye-based polarizing films are being actively studied [1,2,3].

A dye-based polarizing film is created by stretching a dyed poly(vinyl alcohol) (PVA) film in a boric acid solution. This stretching process aligns the dye molecules along the host polymeric chains, and the primary absorption axis of the dye aligns with the stretching direction of the PVA polymer film. As a result, this leads to anisotropic light absorption in the visible wavelength range [4,5,6].

Until recently, only direct dyes, which are adequate for the adsorption and alignment in PVA due to their structural characteristics such as linearity, coplanar conformation, and hydrogen bonding sites, have been used for dye-based polarizing films [7,8]. However, in the case of direct dyes, the primary binding force with PVA is hydrogen bonding, so in very humid conditions, there is concern that the hydrogen bonds between the dye and PVA may weaken, leading to a decrease in durability. Additionally, the process of fabricating PVA polarizing film using direct dyes follows the same mechanism as dyeing cotton fibers with direct dyes. Cotton fibers dyed with direct dyes are known for their poor washing fastness. This means that the hydrogen bonds between direct dye molecules and cotton fibers can be disrupted by water, causing the direct dye to dissolve in water and change the fiber’s color [9]. In order to overcome these issues associated with direct dyes, researchers have explored the use of reactive dyes. As shown in Figure 1, reactive dyes form covalent bonds between the dye and cotton fibers, preventing dye dissolution in water and ensuring high wash fastness for the dyed cotton fibers [10]. These reactive dyes can serve as an alternative material to circumvent problems in the hydrogen bonding system between PVA and direct dyes, particularly in high-humidity conditions. However, reactive dyes, which can be applied to PVA through the covalent bonding of their reactive systems with hydroxyl groups of PVA polymer, have not been studied through comparison with direct dye-based polarizing films yet.

In this study, five reactive dyes were synthesized and used in dye-based polarizing films to assess their potential as novel dichroic materials. The transition moments and alignment of the reactive dyes were compared to those of the corresponding direct dyes. The optical anisotropy of the polarizing films was examined based on the orientational behavior of both reactive and direct dyes. The durability of the prepared polarizing films was evaluated by calculating color differences under high-temperature and -humidity conditions.

## 2. Experimental Section

### 2.1. Materials and Instrumentation

Aniline, 7-amino-4-hydroxy-2-naphthalenesulfonic acid (J acid), 2,4,6-trichloro-1,3,5-triazine (cyanuric chloride) and boric acid purchased from Sigma-Aldrich were used. All other chemicals used in this study were of synthesis grade. Poly(vinyl alcohol) film was supplied by Kuraray Co., Ltd. (Tokyo, Japan) (degree of polymerization 1700, degree of saponification 99.9%, thickness 75 μm). We recorded ^1^H NMR spectra using a Bruker (Billerica, MA, USA) Avance 500 spectrometer operating at 500 MHz. DMSO-*d*_6_ was used as the solvent, and TMS served as the internal standard. FT-IR spectra were obtained with a MIDAC PRS infrared spectrometer using KBr discs. Melting points, which are uncorrected, were determined using a BÜCHI (Flawil, Switzerland) Melting Point B-540 instrument. Absorption spectra were measured using an HP (Palo Alto, CA, USA) 8452A spectrophotometer equipped with a Glan–Thompson polarizer.

### 2.2. Synthesis of Direct Dyes

#### 2.2.1. Synthesis of Dye Intermediate (**I1**)

A solution of 0.69 g (0.01 mol) of sodium nitrite in a small amount of water was added dropwise to a mixture of 0.93 g (0.01 mol) of aniline, 25 mL of water, and 5 mL of 37% hydrochloric acid at a temperature range of 0 to 5 °C. The resulting solution was stirred for 2 h. To scavenge any excess nitrous acid, a small amount of sulfamic acid was used. The diazonium salt solution was then combined with a solution containing 2.39 g (0.01 mol) of the coupling component (J acid) in 200 mL of water at 0~5 °C, maintaining the pH of the solution in the range of 9.5 to 10.5.

After 2 hrs, the solid precipitate underwent filtration, a water rinse, and vacuum oven drying. Next, the raw product was subjected to a 2 h ethanol reflux, followed by a hot filtration, hot ethanol washing, and ultimately, vacuum oven drying (Figure 1).

(*Sodium (E)-7-amino-4-hydroxy-3-(phenyldiazenyl)naphthalene-2-sulfonate,*
**I1**): yield = 76.5%; mp > 300 °C (decomp.); ^1^H NMR (ppm, DMSO-*d*_6_, 500 MHz): δ_H_ 6.35 (2H, s, NH_2_), 6.64 (1H, s, ArH), 6.70 (1H, d, *J* = 8.6 Hz, ArH), 7.15 (1H, t, *J* = 14.7 Hz, ArH), 7.24 (1H, s, ArH), 7.41 (2H, t, *J* = 15.8 Hz, ArH), 7.65 (2H, d, *J* = 7.7 Hz, ArH), 7.93 (1H, d, *J* = 8.6 Hz, ArH), 16.0 (1H, s, OH); FT-IR (KBr, cm^−1^): 1056, 1223 (SO_3_), 1596 (C=O).

#### 2.2.2. Synthesis of Direct Dyes (**D1**~**D5**)

An overnight stirring was applied to a mixture comprising 3.65 g (0.01 mol) of the diazo component (**I1**), 100 mL of water, and 0.69 g (0.01 mol) of sodium nitrite in a small volume of water. At room temperature, 6 mL of 37% hydrochloric acid was slowly added dropwise to the diazo component solution and stirred for 3 h. A minor quantity of sulfamic acid was used as a scavenger for any excess nitrous acid. The resulting diazonium salt solution was combined with a solution containing 2.39 g (0.01 mol) of the coupling component (J acid) in 300 mL of water, maintaining a pH of 8 at 30 °C. After stirring for 2 hrs, the solid precipitate was filtered, washed with water, and dried using a vacuum oven. The raw product, referred to as **D1**, was subjected to a 2 h reflux in ethanol, followed by hot filtration, washing with hot ethanol, and final drying in a vacuum oven (Figure 1). **D2**~**D5** were prepared, and their structures are presented in the previous study [11].

(*Sodium 7-amino-4-hydroxy-3-((E)-(5-hydroxy-6-((E)-phenyldiazenyl)-7-sulfonatonaphthalen-2-yl)diazenyl)naphthalene-2-sulfonate*, **D1**): yield 73.9%; mp > 300 °C (decomp.); ^1^H NMR (ppm, DMSO-*d*_6_, 500 MHz): δ_H_ 6.51 (2H, s, NH_2_), 6.65 (1H, s, ArH), 6.69 (1H, d, *J* = 8.6 Hz, ArH), 7.24 (1H, t, *J* = 14.4 Hz, ArH), 7.28 (1H, s, ArH), 7.47 (2H, t, *J* = 15.3 Hz, ArH), 7.50 (1H, s, ArH), 7.79 (2H, d, *J* = 7.8 Hz, ArH), 7.84 (1H, s, ArH), 7.87 (1H, d, *J* = 7.0 Hz, ArH), 7.94 (1H, d, *J* = 7.1 Hz, ArH), 8.26 (1H, d, *J* = 7.2 Hz, ArH), 15.8 (1H, s, OH), 16.2 (1H, s, OH); FT-IR (KBr, cm^−1^): 1051, 1221 (SO_3_), 1608 (C=O).

### 2.3. Synthesis of Reactive Dyes (***R1***~***R5***)

In this stage, 0.54 g (3 mmol) of cyanuric chloride in 15 mL of acetone was added into a solution of 1.91 g (3 mmol) of disazo direct dye (**D1**) in 100 mL of water at 0~5 °C, and pH was maintained at 5 to 6 via the addition of sodium carbonate solution. Using a salting-out method, we isolated the disazo reactive dye **R1**, subsequently subjecting it to filtration, water rinsing, and vacuum oven drying (Figure 2). The process for preparing disazo reactive dyes **R2** through **R5** closely mirrored that of **R1**. Structures of the reactive dyes **R1**~**R5** are shown in Figure 2. Further information regarding yields, melting points, and ^1^H-NMR and FT-IR data for these dyes is included below.

*(Sodium 7-((4,6-dichloro-1,3,5-triazin-2-yl)amino)-4-hydroxy-3-((E)-(5-hydroxy-6-((E)-phenyldiazenyl)-7-sulfonatonaphthalen-2-yl)diazenyl)naphthalene-2-sulfonate*, **R1**): yield 75.9%; mp > 300 °C (decomp.); ^1^H NMR (ppm, DMSO-*d*_6_, 500 MHz): δ_H_ 7.26 (1H, t, *J* = 14.7 Hz, ArH), 7.48 (1H, t, *J* = 15.7 Hz, ArH), 7.55 (1H, s, ArH), 7.59 (1H, s, ArH), 7.77 (1H, d, *J* = 8.7 Hz, ArH), 7.80 (2H, d, *J* = 7.8 Hz, ArH), 7.97 (1H, s, ArH), 7.99 (1H, s, ArH), 8.01 (1H, d, *J* = 8.7 Hz, ArH), 8.24 (1H, d, *J* = 8.6 Hz, ArH), 8.30 (1H, d, *J* = 8.6 Hz, ArH), 10.9 (1H, s, NH), 15.8 (1H, s, OH), 16.2 (1H, s, OH); FT-IR (KBr, cm^−1^): 772 (C−Cl), 1060, 1217 (SO_3_), 1612 (C=O).

*(Sodium 7-((4,6-dichloro-1,3,5-triazin-2-yl)amino)-4-hydroxy-3-((E)-(5-hydroxy-6-((E)-(4-nitrophenyl)diazenyl)-7-sulfonatonaphthalen-2-yl)diazenyl)naphthalene-2-sulfonate*, **R2***)*: yield 78.8%; mp > 300 °C (decomp.); ^1^H NMR (ppm, DMSO-*d*_6_, 500 MHz): δ_H_ 6.66 (1H, s, ArH), 6.69 (1H, d, *J* = 8.6 Hz, ArH), 7.28 (1H, s, ArH), 7.51 (1H, s, ArH), 7.83 (2H, s, ArH), 7.92 (3H, d, *J* = 8.5 Hz, ArH), 8.21 (1H, d, *J* = 9.3 Hz, ArH), 8.29 (2H, d, *J* = 10.7 Hz, ArH), 10.9 (1H, s, NH), 15.5 (1H, s, OH), 15.7 (1H, s, OH); FT-IR (KBr, cm^−1^): 776 (C−Cl), 1063, 1219 (SO_3_), 1335, 1487 (NO_2_), 1610 (C=O).

*(Sodium 3,3′-((1E,1′E)-((E)-ethene-1,2-diylbis(4,1-phenylene))bis(diazene-2,1-diyl))bis(7-((4,6-dichloro-1,3,5-triazin-2-yl)amino)-4-hydroxynaphthalene-2-sulfonate)*, **R3***)*: yield 83.8%; mp > 300 °C (decomp.); ^1^H NMR (ppm, DMSO-*d*_6_, 500 MHz): δ_H_ 7.28 (2H, s, ArH), 7.32 (2H, d, *J* = 5.2 Hz, ArH), 7.51 (2H, s, −CH=CH−), 7.73 (4H, d, *J* = 6.4 Hz, ArH), 7.78 (4H, d, *J* = 7.8 Hz, ArH), 7.93 (2H, s, ArH), 8.25 (2H, d, *J* = 8.7 Hz, ArH), 10.9 (2H, s, NH), 16.2 (2H, s, OH); FT-IR (KBr, cm^−1^): 770 (C−Cl), 1053, 1190, (SO_3_), 1626 (C=O).

*(Sodium 3,3′-((1E,1′E)-((E)-ethene-1,2-diylbis(4,1-phenylene))bis(diazene-2,1-diyl))bis(6-((4,6-dichloro-1,3,5-triazin-2-yl)amino)-4-hydroxynaphthalene-2-sulfonate)*, **R4***)*: yield 58.1%; mp > 300 °C (decomp.); ^1^H NMR (ppm, DMSO-*d*_6_, 500 MHz): δ_H_ 7.36 (2H, d, *J* = 6.3 Hz, ArH), 7.57 (2H, s, −CH=CH−), 7.73 (2H, s, ArH), 7.75 (2H, d, *J* = 8.2 Hz, ArH), 7.78 (4H, d, *J* = 8.0 Hz, ArH), 7.84 (4H, d, *J* = 8.4 Hz, ArH), 8.61 (2H, s, ArH), 10.8 (2H, s, NH), 16.3 (2H, s, OH); FT-IR (KBr, cm^−1^): 772 (C−Cl), 1053, 1190 (SO_3_), 1623 (C=O).

*(sodium 3,3′-((1E,1′E)-((E)-ethene-1,2-diylbis(4,1-phenylene))bis(diazene-2,1-diyl))bis(8-((4,6-dichloro-1,3,5-triazin-2-yl)amino)-4-hydroxynaphthalene-2-sulfonate)*, **R5**): yield 66.8%; mp > 300 °C (decomp.); ^1^H NMR (ppm, DMSO-*d*_6_, 500 MHz): δ_H_ 7.34 (2H, d, *J* = 7.7 Hz, ArH), 7.56 (2H, t, *J* = 15.4 Hz, ArH), 7.71 (2H, s, −CH=CH−), 7.75 (4H, d, *J* = 7.0 Hz, ArH), 7.82 (4H, d, *J* = 7.6 Hz, ArH), 7.85 (2H, s, ArH), 8.21 (2H, d, *J* = 7.3 Hz, ArH), 10.8 (2H, s, NH), 16.3 (2H, s, OH); FT-IR (KBr, cm^−1^): 766 (C−Cl), 1050, 1194 (SO_3_), 1620 (C=O).

### 2.4. Preparation of Polarizing Films

Dyebath was prepared with each dye (4% o.w.f.) and 0.1 wt% Na_2_SO_4_. For **D1**, a PVA film was immersed in the dyeing solution at 50 °C for 10 min [12,13,14]. For **R1**~**R5**, the PVA film was soaked in the dyeing solution at 50 °C and pH 9 for 10 min. Polarizing films dyed with **D2**~**D5** were prepared as previously described [11]. After dyeing, the PVA films were drawn six times in a 3 wt% boric acid solution at 50 °C. After stretching, the films were washed with water and left to dry at room temperature under consistent tension for one day.

### 2.5. Preparation of Polarizing Films

We assessed the optical anisotropy of the polarizing films with a UV-Vis spectrophotometer that featured a Glan–Thompson polarizer. At the absorption maximum of the polarizing films, the dichroic ratio (R) and order parameter (S) were determined, as described by Equations (1) and (2), respectively. It is worth noting that both the dichroic ratio and order parameters are entirely contingent on the optical properties and alignment of the dyes [15].
R = A_‖_/A_⊥_(1)
S = (R − 1)/(R + 2)(2)
where A_‖_ represents the absorbance of the dyed polymer when illuminated by polarized light vibrating parallel to the stretching direction, while A_⊥_ signifies the absorbance with polarized light perpendicular to stretching.

### 2.6. Molecular Orbital (MO) Calculation

We used the CAChe 6.1.8 software package [16] to optimize the geometry of the dye structures. This optimization involved molecular mechanics using the MM3 force field and iterative energy-minimizing routines employing the conjugate gradient minimizer algorithm [17]. Additionally, the CONFLEX conformational search procedure was applied to discover low-energy conformations of the dye molecules [18]. We also explored the semi-empirical method, PM5, for geometry optimization; however, it was found to be less satisfactory compared to the molecular mechanical approach. Throughout the conformer optimization process, the dye molecules were assumed to have azo linkages, with the central cyanine bridge in the trans configuration.

We determined the directions of the transition dipole moments for the dichroic dyes using the INDO/1 method within the ZINDO package [19], based on the dye structures optimized through the calculations mentioned earlier. A configuration interaction (CI) calculation was performed involving up to 676 configuration functions generated through the single excitation (S-CI) from the ground-state function [20,21]. The CI calculation adapted the occupied orbitals of HOMO−25 to HOMO and the vacant orbitals of LUMO to LUMO+25.

### 2.7. Durability Measurement of Polarizing Films under High Temperature and Humidity

The polarizing films prepared with dichroic dyes were subjected to a Temperature-Humidity Chamber (TH3-E-200, Jeiotech Co., Ltd., Daejeon, Republic of Korea) at 60 °C/93% (R.H.) and 85 °C/85% (R.H.) for 240 h. The chromaticity difference (ΔE_ab_) values were measured using a color spectrophotometer (LCF A2000, Otsuka Electronics Co., Ltd., Osaka, Japan) before and after the exposure to humidity and heat.

## 3. Results and Discussion

### 3.1. Synthesis and Spectral Properties of Direct and Reactive Dyes

2-Azo-1-naphthol derivatives exist predominantly in the hydrazone form via intramolecular hydrogen bonds, which results in the linearity and coplanar conformation of the dyes [11]. The proton peaks of **D1**, which are associated with intramolecular hydrogen bonding, appeared significantly downfield compared to the usual proton peaks of hydroxyl groups. These distinct peaks at 15.8 and 16.2 ppm were confirmed via ^1^H-NMR. Additionally, peaks observed at 1608 cm^−1^ were attributed to the C=O stretching vibration of the dyes, which predominantly exist in the hydrazone form. Reactive dyes have the same structures compared to the corresponding direct dyes, with the exception of the amino group being substituted with cyanuric chloride. The amine proton peaks of the reactive dyes were shifted to a much lower field (10.8~10.9 ppm) due to the substitution of cyanuric chloride. Most of the remaining proton peaks showed a similar trend in comparison with the corresponding direct dyes. The bands due to the C−Cl stretching vibration appeared at 766~776 cm^−1^. The bands of the SO_3_ and C=O stretching vibrations were identified in the FT-IR spectra.

Table 1 presents the spectral properties of reactive dyes and their corresponding direct dyes, which were measured in DMF. The absorption maxima of the reactive dyes were expected to be scarcely different from those of the corresponding direct dyes, but the stilbene-based reactive dyes **R3**~**R5** showed a hypsochromic shift compared to the corresponding direct dyes **D3**~**D5**. The reactive dyes had lower molar extinction coefficients than the corresponding direct dyes, presumably because the substitution of cyanuric chloride disturbs the effective overlap between the aromatic pi cloud of naphthalene ring and the lone pair orbital on the nitrogen atom of the amino group that is substituted with cyanuric chloride.

### 3.2. Comparison Optical Properties between Direct Dye-Based PF and Reactive Dye-Based PF

When dichroic dye molecules are in a stretched PVA film, they are aligned along the drawing direction of the film, as shown in Figure 3. The direction of each dye molecule deviates from the drawing direction of the film **N**, and the order parameter S_M_ and transition moment order parameter S_T_ can be defined by Equations (3) and (4), respectively. Therefore, in order to investigate the orientational behavior of the dye molecules, it is important to know the θ and β value [11,22,23,24]. In this study, the θ and β values were derived by referring to the calculation methods of previous studies [11,25,26,27].
S_M_ = 1/2(3 <cos^2^θ> − 1)(3)
where **M** is the direction of the dye molecular axis.

θ is the angle between **N** and **M**.

<cos^2^θ> is the average of cos^2^θ for all the dye molecules.
S_T_ = 1/2[S_M_(2 − 3sin^2^β)](4)
where **T** is the transition dipole moment of the dye.

β is the angle between **M** and **T**.

The molecular axis **M** for a dye is defined as the longest axis passing through the molecule and encompassing the dye’s π chromophore, as illustrated in Figure 4. To calculate the β values in the most stable conformation of the dichroic dyes, we employed CAChe in combination with the CONFLEX/MM3 method, followed by the ZINDO method.

Table 2 shows various parameters relevant to the optical anisotropy of the direct dye and reactive dye-based polarizing films. The molecular order parameters (S_M_) for both the reactive and direct dyes closely matched the transition moment order parameters (S_T_) because of their small deviation angles (β). As the θ values of the dyes decreased, the dichroic ratios of the films simultaneously increased. This change indicated that aligning the dyes more closely with the drawing direction of the host films improved their optical anisotropy. This effect was linked to the relatively small β values of the dichroic dyes compared to their θ values. In particular, **R3** showed the smallest θ value, indicating that the alignment angle between the PVA and the dye molecule was the lowest, and may increase the order parameter of the dye. On the other hand, in the case of **R4**, although β was the smallest among the prepared reactive dyes, the order parameter value was lower than that of **R3** since θ was larger than **R3**.

Comparing reactive dyes with the corresponding direct dyes, the presence of reactive groups did not significantly affect the θ values of the dyes (Table 2). This suggests that reactive dyes exhibit similar orientational behavior to direct dyes. In a manner similar to direct dyes, which adsorb onto PVA polymer chains through physical interactions such as Van der Waals forces and hydrogen bonding, reactive dyes seem to easily adopt the alignment of the mobile polymer segments along the stretching direction of the host films, whether they have one or two reactive groups. Although the reactive groups exerted a slightly unfavorable effect on the dye alignment, the novel reactive dyes synthesized in this study show potential for use as dichroic materials in polarizing films.

### 3.3. Durability of Polarizing Films under High Temperature and Humidity

From an optical point of view, the durability of a polarizing film depends on the durability of the dye itself and its interaction with PVA. All the direct and reactive dyes synthesized in this study showed high thermal stability, with melting points exceeding 300 °C (Section 2.3). This indicates that the durability of the polarizing film is primarily determined by the bonding strength between the dye and PVA. There is a clear difference in the type of bonding between reactive dyes and PVA, as compared to that between direct dyes and PVA. While hydrogen bonding is the primary interaction between direct dyes and PVA, reactive dyes are bonded to PVA via covalent bonds. Therefore, reactive dye-based polarizing films may be more resistant to heat and moisture than direct dye-based polarizing films.

For commercial display applications, colored optical parts such as color filters, polarizing films, and protective films should have a chromaticity difference (ΔE_ab_) value of less than 3 after 240 h of testing under specified conditions [28,29]. Figure 5 shows the color difference (ΔE_ab_) value of direct dye and reactive dye-based polarizing films under high temperature and humidity.

The direct dye (**D3**)-based polarizing film showed a color difference of less than 1 after 72 h (3 days) under 60 °C/93% (R.H.) conditions, and a commercially usable level of 1.76 after 240 h. However, under the more severe conditions of 85 °C/85% (R.H.), the color difference reached 2.73 after 240 h, which may not be suitable for commercial use. In contrast, the reactive dye (**R3**)-based polarizing film showed a color difference of less than 1 after 240 h 60 °C/93% (R.H.) conditions, and it was difficult to distinguish with the naked eye. Additionally, the color difference of the **R3**-based polarizing film was 1.14 even after 240 h under the 85 °C/85% (R.H.) condition, demonstrating excellent durability under extremely high temperature and humidity.

The durability for the **D4**- and **R4**-based polarizing film was also evaluated (Figure 5b). The **R4**-based polarizing film demonstrated a durability value similar to that of the **R3**-based polarizing film. When comparing the difference in durability with the **D4**-based polarizing film, a counter direct dye for **R4**, the results resembled the differences in durability observed between **R3**-based and **D3**-based polarizing films. So, it can be inferred that reactive dye-based polarizing films are more stable in high-temperature and -humidity environments compared to direct dye-based polarizing films due to the covalent bonds between the reactive dyes and PVA. As shown in Figure 1, the hydrogen bond between the direct dye and the PVA can be easily broken by water molecules in high-humidity conditions. On the contrary, the strong covalent bonds between the reactive dye and PVA ensure stability in harsh environments, as the dye can be hardly affected by moisture.

## 4. Conclusions

Reactive dyes as novel dichroic materials were synthesized and introduced in poly(vinyl alcohol) polarizing films. We analyzed the orientational characteristics of the dyes and the optical anisotropy of the polarizing films by evaluating their transition moments and making comparisons with the corresponding direct dyes. The transition dipole moments of the reactive dyes exhibited minimal deviation from their molecular axes, much like the behavior observed in the case of the direct dyes. Due to this minor change in β values to θ values, the dichroic ratios of the films primarily depended on the alignment of dyes towards their drawing direction. The orientational behavior of the reactive dyes was similar to that of the direct dyes irrespective of the number of their reactive group. In particular, in the case of reactive dyes, due to the strong covalent bond between the PVA and dye, the color difference was reduced by more than 50% compared to direct dyes under high-temperature and humid conditions. Therefore, the novel reactive dyes synthesized in this study have the potential to be more useful for industrial applications requiring high-temperature and -humidity resistance.

## Data Availability

Data are contained within the article.

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
