# Peer review of "Improving Durability of Dye-Based Polarizing Films Using Novel Reactive Dyes as Dichroic Materials"

_polymers, 2023, doi:10.3390/polym15224365_

Round 1

Reviewer 1 Report

Comments and Suggestions for Authors

Is dye-based PF widely put in production? Will reactive dye-based PF cause difficulties or cost more in production?

In durability test, more sample could be tested for precious verify. e.g. D1 versus R1, D2 versus R2.

Reviewer 2 Report

Comments and Suggestions for Authors

The present work reports the synthesis of five reactive dyes and their use for the preparation of polarized PVA films, as compared with the respective behavior of PVA films prepared using direct dyes. The main outcome is that the durability of polarized films using reactive dyes is significantly improved as compared to the use of direct dyes.

The manuscript is of interest and could be accepted for publication. However, a major revision s probably necessary (to improve the clarity and completeness of presentation), as follows:

i)                 Introduction section is clear. However, it could be more detailed (for example, a: be more detailed as it concerns the use of direct dyes and b: discuss the reaction mechanism and conditions concerning dyeing of PVA with reactive dyes).  Figure 5 could be used in this introductory section, also.

ii)                References section. References are very few and old. The authors should try to update the references reported.

iii)              Section 3.1. The synthesis and characterization are discussed in this section, based on NMR, FTIR and UV-vis studies. It would be very helpful for the reader if the respective spectra were presented, at least as supplementary information.

iv)              Section 2.4. The authors mention here that “For R1~R5, the PVA film was soaked in the dyeing solution at 50 °C and pH 9 for 10 min.” As mentioned, a literature review on the use of reactive dyes and conditions for dyeing PVA (or similar substrates)  should be done.  Are these conditions sufficient for covalent bonding? Is there any evidence (apart from durability tests) that covalent bonding is indeed achieved?

v)                A discussion on the preparation and characterization of PVA films is missing. This could include the appearance of the films (photos, maybe), dye uptake (if possible) chromaticity values, etc.

Comments on the Quality of English Language

Dear Editor,

I find that the quality of English is rather good. Minor mistakes can be corrected during proof-reading process, if the manuscript will be accepted for publication.

Best Regards

Georgios Bokias

Round 2

Reviewer 2 Report

Comments and Suggestions for Authors

The authors have tried to make changes or respond satisfactorily  to all questions raised.

The manuscript can be accepted for publication.